# Modulation of perovskite degradation with multiple-barrier for light-heat stable perovskite solar cells

Jing Zhou[1,6], Zonghao Liu [1,2,6] ✉, Peng Yu[1,3,6], Guoqing Tong[4,6], Ruijun Chen[5], Luis K. Ono [4], Rui Chen[1], Haixin Wang[1], Fumeng Ren[1], Sanwan Liu[1], Jianan Wang[1], Zhigao Lan[5], Yabing Qi [4] ✉ & Wei Chen [1,2] ✉

The long-term stability of perovskite solar cells remains one of the most important challenges for the commercialization of this emerging photovoltaic technology. Here, we adopt a non-noble metal/metal oxide/polymer multiple-barrier to suppress the halide consumption and gaseous perovskite decomposition products release with the chemically inert bismuth electrode and $Al_2O_3$/parylene thin-film encapsulation, as well as the tightly closed system created by the multiple-barrier to jointly suppress the degradation of perovskite solar cells, allowing the corresponding decomposition reactions to reach benign equilibria. The resulting encapsulated formamidinium cesium-based perovskite solar cells with multiple-barrier maintain 90% of their initial efficiencies after continuous operation at 45 °C for 5200 h and 93% of their initial efficiency after continuous operation at 75 °C for 1000 h under 1 sun equivalent white-light LED illumination.

The stability issue of perovskite solar cells (PSCs) has been recognized as one of the major bottlenecks impeding their commercialization[1–3]. For practical application, passing stability test protocols, such as the International Electrotechnical Commission (IEC) 61215:2016 norm is an essential requirement for their practical application. Protocols such as thermal cycling, damp heat, humidity freeze, mechanical load, hail test, ultraviolet light, and outdoor test in IEC 61215:2016 norm are mainly used to evaluate the mechanical stability and the encapsulation quality, making this norm mainly suitable for solar cells using intrinsic stable active materials under light[4]. Different from silicon, perovskites could decompose under light, so it is important to include the light aging factor in the protocols to evaluate the stability of PSCs[4,5]. Since the light-induced degradation will be significantly accelerated together with elevated temperature[6], light and elevated temperature-induced degradation (LeTID) has been recognized to be one of the

harshest stability tests for PSCs in the perovskite photovoltaic community[5,7–9].

To improve the operational stability of PSCs at elevated temperatures, formamidinium cesium (FACs) perovskites have been used to replace methylammonium lead iodide in earlier stage due to their larger enthalpy and activation energy for the decomposition reactions[10]. However, FA-based perovskites could still undergo decomposition under light and elevated temperature[6,10]. Fortunately, the decomposition reactions (I-IV) at temperatures below 95 °C are reversible. The irreversible reaction (V) only occurs at temperatures above 95 °C, which is well above the usually considered operational temperature range of solar cells[10]:

$$CH(N_2H_4)PbI_3 \rightleftharpoons CH(N_2H_4)I + PbI_2 \qquad (1)$$

[1]Wuhan National Laboratory for Optoelectronics (WNLO), Huazhong University of Science and Technology (HUST), 430074 Wuhan, China. [2]Optics Valley Laboratory, Wuhan, Hubei 430074, China. [3]China-EU Institute for Clean and Renewable Energy, Huazhong University of Science and Technology, 430074 Wuhan, Hubei, China. [4]Energy Materials and Surface Sciences Unit (EMSSU), Okinawa Institutes of Science and Technology Graduate University (OIST), 1919-1 Tancha, Onna-son, Kunigami-gun, Okinawa 904-0495, Japan. [5]School of Physics and Telecommunications, Huanggang Normal University, 438000 Huanggang, Hubei, China. [6]These authors contributed equally: Jing Zhou, Zonghao Liu, Peng Yu, Guoqing Tong. ✉e-mail: liuzonghao@hust.edu.cn; Yabing.Qi@OIST.jp; wnlochenwei@hust.edu.cn

$$CH(N_2H_4)I \rightleftharpoons CH(N_2H_3) + HI \qquad (2)$$

$$PbI_2 \rightleftharpoons Pb^0 + I_2 \qquad (3)$$

$$CH(N_2H_3) \rightleftharpoons HCN + NH_3 \qquad (4)$$

$$3CH(N_2H_3) \rightarrow C_3H_3N_3 + 3NH_3 \qquad (5)$$

The suppression of the outgassing and consumption of these decomposition products could enable these decomposition reactions (1–4) to reach benign equilibria and prolong the life span of PSCs from the viewpoint of reaction thermodynamics. Thus, encapsulating PSC into a tight enclosure is highly desirable[6,11,12].

To date, adhesive-covered glass encapsulation, thin-film barrier strategy or the combination of thin-film barrier strategy and cover glass encapsulation have been demonstrated to be effective in suppressing the decomposition of perovskite for device stability improvement[6,12–19]. For example, Shi et al. used a pressure-tight polymer/glass blanket encapsulation strategy to suppress the decomposition of perovskite components and an Au electrode to inhibit the consumption of perovskite components, ultimately achieving no efficiency decay after 1800 h of damp-heat aging[6]. In terms of thin-film barrier, we previously employed sol-gel Ti(Nb)O$_x$[20] and CeO$_x$[21] electron-extraction layer underneath Ag electrode in p-i-n PSCs to reduce electrode corrosion and protect the perovskite against moisture from the ambient atmosphere. However, sol-gel-derived layers usually are relatively poor permeation barriers when compared with atomic layer deposition (ALD)-based layers. Riedle et al. introduced an AZO/ atomic layer deposition (ALD)-grown SnO$_2$ bilayer electron-extraction interlayer below the Ag electrode in p-i-n PSCs[16]. The AZO/SnO$_2$ bilayer barrier effectively hinders the ingress of moisture, the egress of perovskite decomposition products and the corrosion of metal electrode, leading to a $T_{60}$ of 300 h under light-heat stability at 60 °C. Similarly, Seo et al. introduced a dense and impermeable ALD-grown AZO as a barrier between electron transport materials and Al or Ag electrode in p-i-n PSCs and achieved a device operational lifetime of $T_{86.7}$ = 500 h at the maximum power point (MPP) under 1 sun illumination at 85 °C for 500 h[22]. We also incorporated ALD-Al$_2$O$_3$ as a barrier atop the metal electrode in a parallel perovskite solar module and achieved $T_{95}$ = 1187 h with MPP tracking under 1 sun illumination at 50 °C[23]. Indeed, the deposition rate of commonly used ALD system in a lab is relatively slow; spatial ALD with a fast deposition rate and good scalability that is widely used in the industrial production line of crystalline Si solar cells is very promising for the industrial manufacture of PSCs. In addition to a single thin-film barrier, it is highly desirable to combine the ALD/CVD technique to seal the perovskite solar cells into a tight closure with multiple thin-film barriers[12]. Especially the excellent barrier properties of chemical vapor deposition grown polymer/ALD-oxide bilayer that have been widely used in organic light emitting diode, electronics, etc. semiconductor products for encapsulation owing to the low gas penetration rate[24,25]. When further combined with cover encapsulation, a robust multiple oxide/polymer thin-film barrier-cover encapsulation strategy is very promising to form a tight system to suppress PSCs' degradation.

Regarding the consumption of perovskites decomposition products within the system created by encapsulation, commonly used metal rear electrodes such as silver (Ag)[26,27], aluminum (Al)[28] and copper (Cu)[29] electrodes could be the culprit due to their tendency to react with halide anions, especially under light and heating conditions. Such consumption largely increases the risk of breaking the thermodynamics equilibriums of perovskite decomposition reactions and leads to the deterioration of device performance. Although PSCs based on gold (Au)[30] or Cr/Au[31] electrodes have achieved decent operational

stability at elevated temperatures, the high price of Au and its tendency to diffuse into perovskite forming antisite defects largely limits its massive deployment for practical application[27]. Transparent conductive oxide (TCO) electrode, such as ITO, is also used as both a barrier and inert rear electrode for stable PSCs[13,14], but the problem is that TCO is expensive and TCO rear electrode could lead to high optical loss and thus cause a lower current density when compared to metal electrode, which hinders the further improvement of device efficiency and reduction of its cost for massive application. On the other hand, the TCO rear electrode deposited on the top of the perovskite and charge transport layers cannot undergo high-temperature annealing to ensure high conductivity due to the sensitivity of perovskite to high temperatures. Thus, the TCO rear electrode usually has a high sheet resistance, which leads to a low fill factor, especially for solar modules[32]. It is thus highly desirable to adopt a low-cost and intrinsic stable metal electrode showing good compatibility with scalable deposition methods to minimize the consumption reactions. From the viewpoint of thermodynamics, a low-cost and robust intrinsic stable electrode/encapsulation barrier design is essential to suppress the outgassing and consumption of the decomposition products for improving PSCs' stability, especially targeting the toughest LeTID.

Herein, we propose the use of intrinsic stable metal electrode/metal oxide/polymer bismuth (Bi)/Al$_2$O$_3$/parylene as multiple-barrier to modulate the decomposition of FACs perovskite in p-i-n PSCs as shown in Fig. 1. The Al$_2$O$_3$/parylene bilayer with a water vapor transmission rate (WVTR) as low as $10^{-5}$ g m$^{-2}$ day$^{-1}$ effectively acts as barrier to block the outgassing of perovskite decomposition products but also suppress the ingress of water and oxygen to suppress perovskite decomposition. In terms of the electrode, we extend our discovery of the chemically inert bismuth (Bi) as a 20-nm-thick interfacial layer to a 1-µm-thick back contact electrode by optimizing its deposition process[29]. The 1-µm-thick Bi electrode not only acts as an effective conductive back contact electrode to enable promising power conversion efficiency (PCE) of 22.26% with an active area of 0.09 cm$^2$ and a PCE of 20.56% with an active area of 1 cm$^2$, but also does not consume perovskite decomposition products in the above reactions (1–4), and further acts as a robust chemical barrier to block the outgassing of perovskite decomposition products. With further cover encapsulation, our FACs perovskite-based p-i-n devices with multiple-barrier maintained 90% of their initial efficiencies after continuous operation at 45 °C for 5200 h and 93% of their initial efficiency after continuous operation at 75 °C for 1000 h under 1 sun equivalent white-light LED illumination.

## Results and discussion

### Modulation of perovskite decomposition with metal oxide/polymer bilayer

Metal oxide and polymer-based thin-film encapsulation is a widely used encapsulation method in electronic devices[12]. We have used a single Al$_2$O$_3$ layer[23] and parylene[11] layer to improve the stability of perovskite solar modules. Here, we integrated two kinds of films together as an Al$_2$O$_3$ (30 nm)/parylene (1 µm) bilayer barrier to encapsulate PSCs due to bilayer could largely increase the limitation of a single layer on blocking capability (Supplementary Table 1)[15]. It should be noted that the bilayer does not show an obvious detrimental effect on the perovskite film quality and device performance, and the device's performance also shows good reproducibility (Supplementary Figs. 1 and 2). The water resistance of the bilayer was proved in Supplementary Note 1, Supplementary Fig. 3 and Supplementary Movie 1.

To further investigate the modulation of perovskite degradation with Al$_2$O$_3$/parylene bilayer under elevated temperatures, we measured the X-Ray diffraction (XRD) evolution of the FACs perovskite films with/without bilayer encapsulation under an in situ heating process. As shown in Fig. 2a, the peaks of PbI$_2$ in the bare perovskite

film appeared at about 160 °C, and the sample almost completely decomposed into $PbI_2$ after the temperature reached about 240 °C. By contrast, the perovskite film with $Al_2O_3$/parylene bilayer barrier started to decompose until the temperature increased to 250 °C. Most importantly, the main peaks of the perovskite film were still retained even when the heating temperature was increased to 280 °C (Fig. 2b). These results indicate that the decomposition reaction of the perovskite film under elevated temperatures is effectively suppressed by the $Al_2O_3$/parylene barrier. The photoluminescence (PL)-mapping evolutions of perovskite films under 170 °C further confirm the effectiveness of ALD-$Al_2O_3$/CVD-parylene bilayer for the suppression of perovskite decomposition (Supplementary Note 2 and Supplementary Fig. 4). In addition, the perfect barrier effect of the bilayer barrier under 85 °C/85% hydrothermal condition and 1-sun light/85 °C LeTID condition have been further confirmed in the comparison studies as shown in Supplementary Note 3 and Supplementary Figs. 5–7.

Above all observations consistently demonstrate that the $Al_2O_3$/parylene bilayer film is a powerful barrier to block the release of decomposed gases to modulate the degradation of perovskite under serious light/heat aging conditions. To further verify this point, we used in situ mass (MS) spectra coupled with a Xenon lamp solar simulator to track the outgassing of perovskite decomposition gaseous products (Supplementary Fig. 8 and Supplementary Movie 2). Figure 2c summarizes the mass-to-charge ratio (m/q) peaks recorded simultaneously during the light accompanied with thermal degradation of the bare perovskite film, and the temperature was recorded with the light applied. As shown in Fig. 2c, FA, HI, HCN and $NH_3$ were found to be the major decomposition products from $FA_{0.85}Cs_{0.15}Pb(I_{0.95}Br_{0.05})_3$ within 90 min test. These findings are consistent with the in situ XRD results (Fig. 2a). For the perovskite film encapsulated with $Al_2O_3$/parylene bilayer (Fig. 2d), even if we increased the light intensity constantly until the temperature of the test chamber arrived at 95 °C (Fig. 2d), where the whole process lasted 1000 h, there was still no signal of degradation product was found. These findings illustrate that the release of degradation products was confined within the compact enclosure created by the $Al_2O_3$/parylene bilayer. Combining the in situ XRD characteristic, the degradation reaction of the perovskite under light is suppressed by the bilayer barrier. For reference, we also checked the samples encapsulated with a single $Al_2O_3$

barrier and a single parylene barrier, respectively (Supplementary Fig. 9). It showed the single layer was not robust enough to prevent the release of the perovskite degradation products. This further confirms that the $Al_2O_3$/parylene bilayer barrier is much more efficient in blocking gas release when compared with organic or inorganic single-layer barrier. In the above, the investigation on the suppression of perovskite decomposition under the stress of humidity, heat, humidity/heat, and light/heat conditions coincidently verifies the robust barrier capability of the $Al_2O_3$/parylene bilayer barrier.

### Modulation of perovskite decomposition with intrinsic stable Bi electrode

In the enclosed space protected by $Al_2O_3$/parylene bilayer, we further used low-cost and chemically inert Bi as the electrode material in p-i-n PSCs to modulate perovskite decomposition. According to our previous research[29], Bi is a chemically inert element with halide perovskites. We previously used Bi as an interfacial barrier to prevent the migration of halide ions and diffusion of Ag. However, the active metals such as Ag atop the Bi barrier can still consume perovskite decomposition components and cause perovskite decomposition within the system created by encapsulation. This is because the metal or halide ion can still interpenetrate and react to cause degradation along with the long-term operation of PSCs. This inspires us to replace the active electrode with inert Bi as the electrode. However, the problem is that the thermally evaporated Bi electrode does not show enough conductivity. Previously, we have found that when the thickness of Bi film deposited by thermal evaporation (T-Bi) was over 40 nm, protuberant columnar Bi crystals with poor connection were formed, and the surface roughness of the T-Bi film increased, resulting in poor device performance[29]. Here, we first tried to improve the conductivity by increasing the thickness of the T-Bi film to 1 μm. However, it is found that the 1-μm-thick T-Bi film has a loose columnar structure (Supplementary Fig. 10a), and the resultant device shows low performance mainly due to the low fill factor (Supplementary Fig. 10b) caused by the high sheet resistance of 1-μm-thick T-Bi electrode (Supplementary Table 2). To reduce the sheet resistance of the T-Bi electrode, we further tried to use the magnetron sputtering method to deposit a dense Bi film (M-Bi) as the electrode. The resultant 1-μm-thick M-Bi film on a glass substrate showed a low sheet resistance of 1.4 Ω

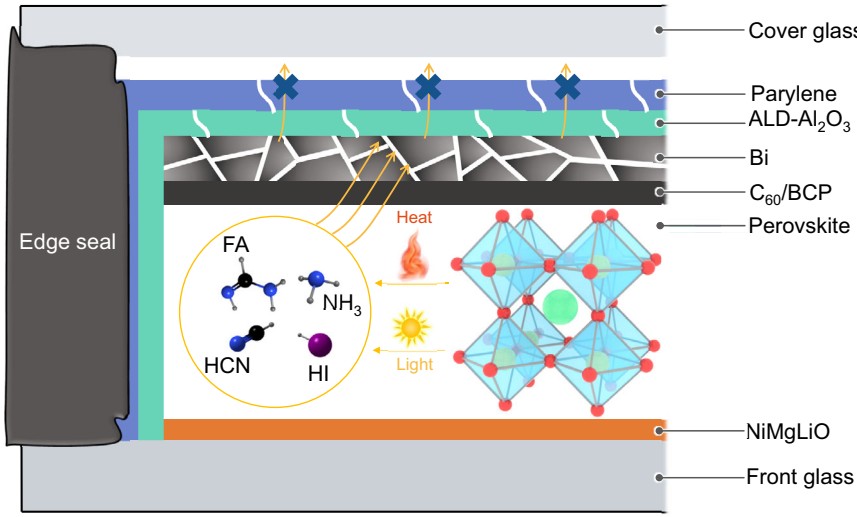

**Fig. 1 | Schematic of the multiple-barrier strategy for PSCs.** The multiple-barrier strategy consists of inert Bi metal electrode and $Al_2O_3$/parylene thin-film encapsulation.

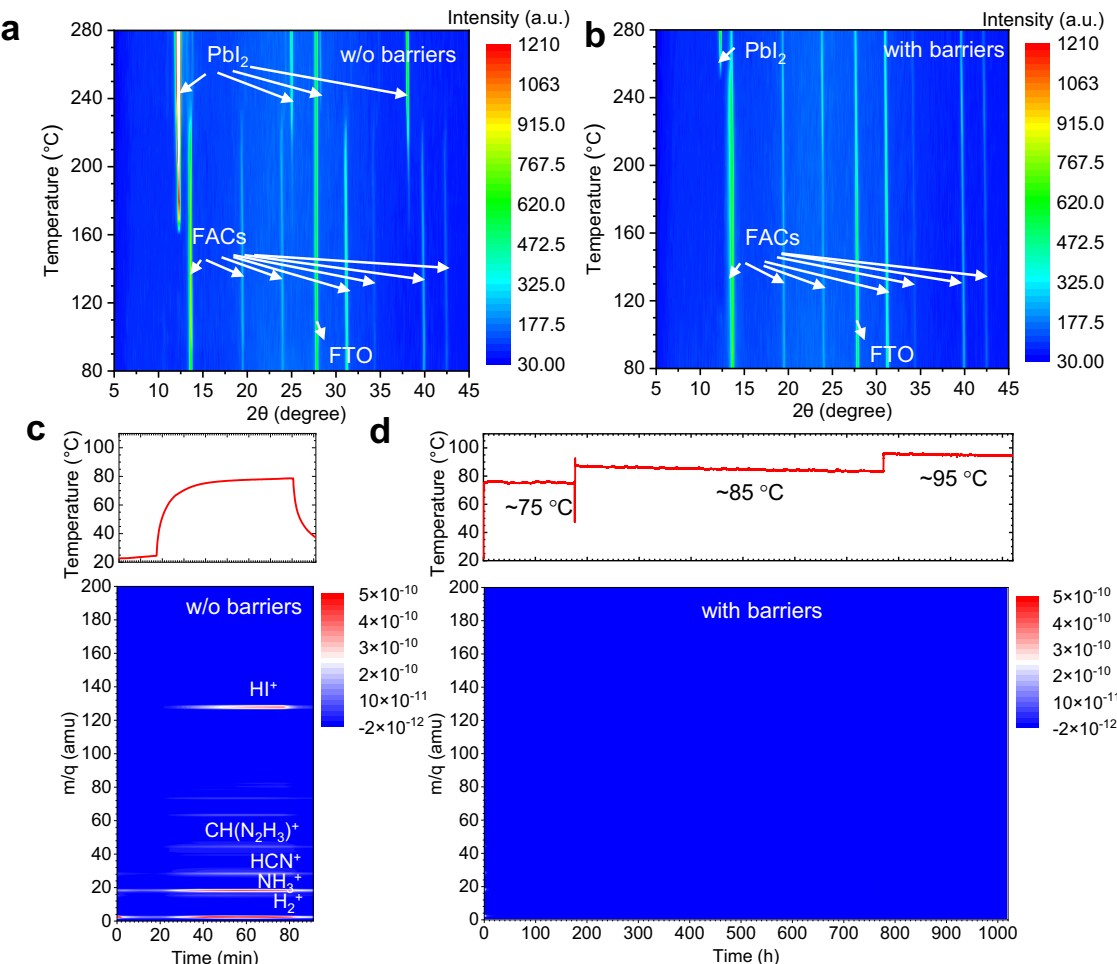

**Fig. 2 | Signature decomposition products of FA-based perovskite films with/without barriers. a, b** In situ heating-XRD characterization of the perovskite films with barrier and without barrier. **c, d** In situ illumination with temperature monitoring of MS characterization of the perovskite films with barrier and without barrier. It is noted that the chemical traces of perovskite films with barriers are below the sensitivity of MS. Source data are provided as a Source Data file.

square$^{-1}$, which is comparable with that of Ag with a thickness of 100 nm. This improvement is due to the much denser structure of M-Bi when compared with T-Bi (Supplementary Fig. 11), and a more compact structure is beneficial for the subsequent deposition of ALD/CVD thin films, ensuring the formation of a more tightly packed barrier layer. The magnetron sputtering technology can increase the electrode deposition rate and utilization rate of raw materials, making it beneficial for industrial applications. However, the high energy irradiation during sputtering could easily destroy perovskite and electron transport layer due to the soft nature of these materials, leading to poor device performance (Supplementary Fig. 12). To solve this problem, we first deposited a 20-nm-thick Bi film by thermal evaporation (T-Bi) and then used magnetron sputtering to deposit 1-μm-thick Bi film on the top of T-Bi film inspired by a previous report[33]. It should be noted that the pre-deposited 20 nm T-Bi film not only helps optimize the morphology of the T-Bi/M-Bi electrode but also acts as a barrier layer to protect the underneath perovskite layer and charge transport layer that are sensitive to high energy. As a result, the obtained T-Bi/M-Bi film also showed a low sheet resistance of about 1.4 Ω square$^{-1}$, and more importantly, the perovskite and charge transport underlayers are not damaged.

We further used T-Bi/M-Bi film as electrode in inverted PSCs based on FACs perovskite. The corresponding cross-sectional scanning electron microscopy (SEM) image of an FACs perovskite-based p-i-n PSCs with an architecture of FTO/NiMgLiO/Perovskite/LiF/C$_{60}$/BCP/Bi/Al$_2$O$_3$/parylene is shown in Fig. 3a. It is found that the Bi electrode is compact and dense, and the device is tightly covered by the Al$_2$O$_3$/parylene bilayer. The fabricated device with an active area of 0.09 cm$^2$ gave a PCE over 22% (22.04%), with an open-circuit voltage ($V_{OC}$) of 1.118 V, a short-circuit photocurrent density ($J_{SC}$) of 24.08 mA cm$^{-2}$, and a fill factor (*FF*) of 0.819 for the forward scan; a PCE of 22.26% with a $V_{OC}$ of 1.119 V, a $J_{SC}$ of 24.09 mA cm$^{-2}$, and an *FF* of 0.826 for the reverse scan under AM 1.5 G illumination (Fig. 3b). For the device with an active area of 1 cm$^2$, a champion efficiency of 20.56% with a $V_{OC}$ of 1.116 V, a $J_{SC}$ of 23.81 mA cm$^{-2}$, and an *FF* of 0.774 was achieved, as shown in Fig. 3c. The corresponding EQE of the device is shown in Supplementary Fig. 13. The EQE remains ≈90% in the wavelength range from 400 to 750 nm, which indicates the effective light harvesting, charge generation and collection. The corresponding integrated current density from the EQE result is 23.39 mA cm$^{-2}$, which is consistent with the value of $J_{SC}$ obtained from the *J-V* test. A stabilized PCE of 20.45% and a stabilized $V_{OC}$ of 1.115 V were also obtained (Fig. 3d). In addition, the PCE of 0.09 cm$^2$ Bi-based device is comparable to the performance of the Ag-based device (Supplementary Fig. 14), which indicates the T-Bi/M-Bi film is a promising electrode for high-efficiency PSCs. It is noteworthy that the efficiency of the FTO/NiMgLiO substrate-based device is 1–2% lower than the ITO/poly(triarylamine) substrate-based one in our lab (Supplementary Fig. 15)[34], but we confirm that the stability for the former one is much better, especially under the harsh LeTID condition (Supplementary Fig. 16).

To study the perovskite decomposition behavior with the existence of different electrodes, in situ heating-XRD characterization was

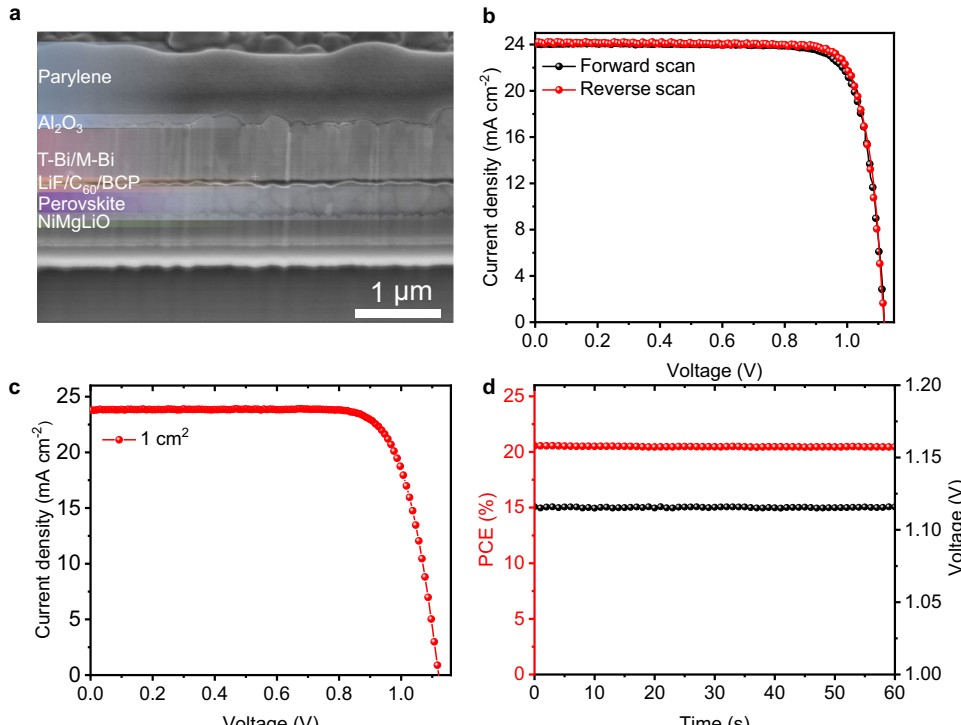

**Fig. 3 | Solar cells photovoltaic performance. a** SEM cross-sectional image of the device (scale bar: 1 μm). **b** *J-V* curves a typical small-area (active area, 0.09 cm²) PSC with a T-Bi/M-Bi electrode based on FACs perovskite. **c** The corresponding *J-V* curve of encapsulated device with an active area of 1 cm². **d** Stabilized PCE and $V_{OC}$ of the device with an active area of 1 cm². Source data are provided as a Source Data file.

carried out on the samples of perovskite/electrode/$Al_2O_3$/parylene cases with Ag and Bi electrodes, respectively. Here, $Al_2O_3$/parylene bilayer barrier was used to minimize the influence of the outgassing of perovskite decomposition products and ingress of $H_2O/O_2$. As shown in Fig. 4 and Supplementary Figs. 17 and 18, the appearance of the $PbI_2$ peaks in the Ag-based sample was at 170 °C, which is much earlier than that of the Bi sample, in which $PbI_2$ peaks were observed at 250 °C. The perovskite crystal structure of the Bi-based sample remained relatively stable when compared with the Ag-based sample. The Ag-based sample also showed an obvious peak with a broader feature near the (101) crystal orientation of about 14°, which may be induced by the lattice mismatch due to perovskite component loss during aging[35]. Moreover, an AgI peak was also observed in the former sample, while no $BiI_3$ peak was observed in the latter one, which indicated that Ag could induce perovskite decomposition accompanied by the formation of AgI, whereas Bi will not react with perovskite. Considering that the sample of perovskite film with $Al_2O_3$/parylene barrier but no electrode starts to decompose until the temperature increases to 250 °C, which is similar to the Bi case. These results confirmed that Bi is an inert metal that will not induce perovskite decomposition. In addition, the relatively lower $PbI_2$ peak intensity for the Ag-based sample than that of the Bi-based sample at temperatures higher than 250 °C suggested that the iodine element in $PbI_2$ could be further consumed by the Ag electrode in the Ag-based sample. The optical images and XRD patterns of perovskite/electrode (Bi and Ag, Cu, and Au) samples under 1 sun light soaking at 85 °C further confirmed the effectiveness of the Bi electrode in modulating perovskite decomposition (Supplementary Figs. 19a and 20). After further being heated to 175 °C in the air for a period of time, the Cu and Ag strip electrodes have caused the underlying perovskite thin film to decompose and turn yellow, while the perovskite thin film under the Bi strip electrode remains unchanged in color. This intuitively demonstrates that the corrosion of metals accelerates the decomposition of perovskite (Supplementary Fig. 19b).

## LeTID stability test of the devices

As demonstrated above, $Al_2O_3$/parylene bilayer barrier and Bi electrode are effective to retard the degradation of perovskite. To further study the influence of the multiple-barrier on the device stability, especially against LeTID, FACs-based p-i-n PSCs with barriers and cover encapsulation including Ag electrode without $Al_2O_3$/parylene bilayer barrier but with Bynel cover encapsulation (Ag-Bynel), Bi electrode-based device without $Al_2O_3$/parylene barrier but with Bynel cover encapsulation (Bi-Bynel), and Bi electrode-based device with $Al_2O_3$/parylene bilayer barrier and Bynel cover encapsulation (Bi-Barrier-Bynel) were fabricated. Bynel cover glass and epoxy-based border encapsulation were used to further protect the devices (Supplementary Fig. 21). First, three kinds of devices were aged under continuous 1 sun equivalent white-light LED illumination with MPP tracking at 45 °C. The stability evolutions of devices are shown in Supplementary Fig. 22. All three devices could maintain high percentages of their initial PCEs under continuous illumination at 45 °C. The Ag-Bynel device maintained 84% of its maximum power output after 1000 h with a degradation rate of −0.016% h⁻¹ [36,37]. The Bi-Bynel device maintained 90% of its maximum power output after 1000 h with a degradation rate of −0.010% h⁻¹. The Bi-Barrier-Bynel device maintained 90% of its maximum power output after 5200 h with a degradation rate of −0.002% h⁻¹, which indicates that the multiple-barrier strategy is effective in improving device stability.

To further validate the effectiveness of multiple-barrier, we employed a combination of 75 °C heating stress and light stressor test to evaluate the stability of the devices, in which the Ag-Bynel device degraded rapidly to failure with a degradation rate of −4.29% h⁻¹ (Fig. 5a). This observation suggested that light-heat stress could trigger the degradation of the device. This is because light-heat stress can considerably accelerate the degradation of perovskite; the use of Ag also consumed a large amount of volatile perovskite decomposition products, leading to an irreversible decomposition of the perovskite layer. When Ag is replaced by Bi, the Bi-Bynel device showed improved

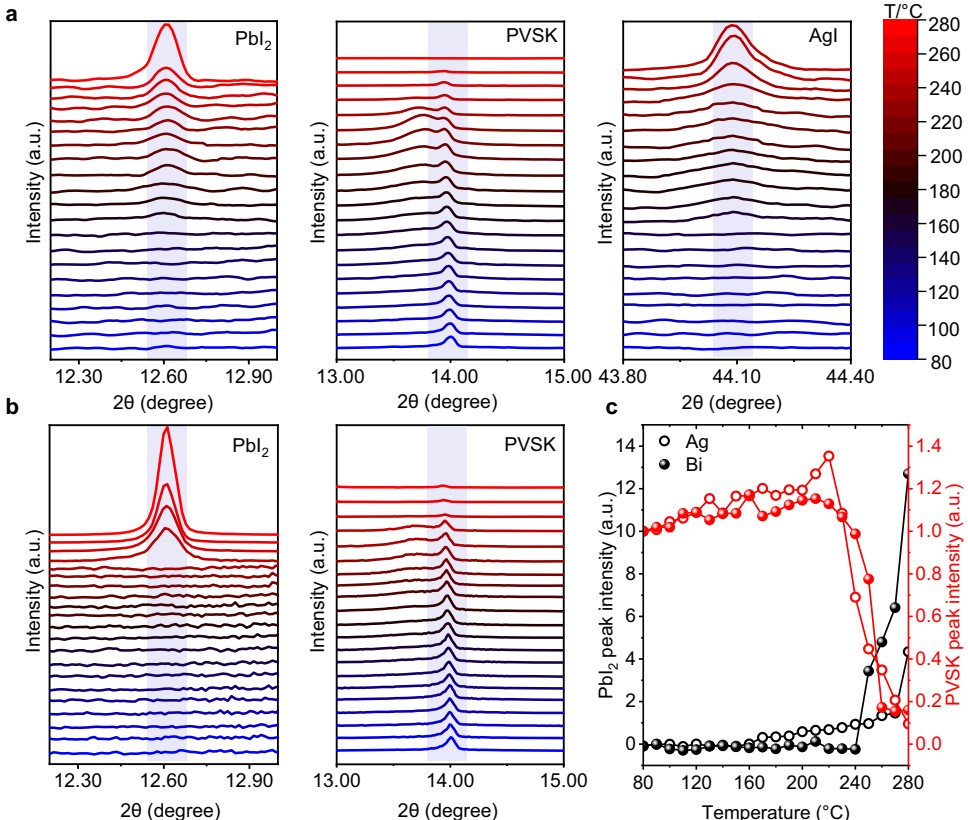

**Fig. 4 | Electrode-induced perovskite decomposition. a** In situ heating-XRD characterization of the FACs/Ag/ALD-Al$_2$O$_3$/CVD-parylene sample and the corresponding detailed PbI$_2$, perovskite (PVSK), AgI pattern, respectively. **b** In situ heating-XRD characterization of the FACs/Bi/ALD-Al$_2$O$_3$/CVD-parylene sample and the corresponding detailed PbI$_2$, PVSK pattern, respectively. **c** Dependence of peak intensity of PbI$_2$ and PVSK on temperature. Source data are provided as a Source Data file.

stability with a linear decay rate of −0.14% h$^{-1}$ during light-75 °C heat aging. This is because Bi is a kind of chemical insert metal, which will not react with the gaseous perovskite degradation products. However, the escape of gaseous perovskite degradation products could still break the chemical equilibriums and accelerate the decomposition of the perovskite absorber in the Bi-Bynel device. When Bi-Barrier-Bynel was used, the corresponding device showed a much slower decay rate of −0.007% h$^{-1}$. This device could maintain 93% of its initial efficiency after 1000 h soaking at 75 °C, which is one of the best light-heat stability to date (Supplementary Table 3). The evolution of all the parameters, including $J_{SC}$, $V_{OC}$, $FF$ and PCE, is shown in Fig. 5b. The $FF$ and $V_{OC}$ are relatively stable, and the decay of PCE mainly comes from the decrease of $J_{SC}$, which is likely due to the slight degradation of perovskite. Improving the stability of perovskite films with additive engineering and dimensional engineering is proposed to further enhance the LeTID stability of PSCs[38,39]. It should be noted that a water-cooling integrated white-light LED light source was used for the stability test because of its more stable spectra and light intensity output with a low degradation rate of <2% within 10,000 h when compared with the commonly used Xenon lamp (normally possesses a degradation rate of ~10% within only 1000 h) in $J$-$V$ measurements. In addition, our multiple-barrier-based devices also showed good stability repeatability (Supplementary Fig. 23) and good scalability (Supplementary Movie 3 and Supplementary Note 4), making the multiple-barrier strategy promising for the application in commercialized large-area solar module production.

To further reveal the influence of multiple-barrier on device stability, cross-sectional SEM and element distribution profiles were measured to visually examine the morphology change of the devices after aging. Obvious voids and morphology change are observed in the

perovskite layer of the Ag-Bynel device, suggesting that the perovskite layer is one of the most important bottlenecks for PSCs due to the LeTID (Supplementary Fig. 24). This observation is consistent with the corresponding operation stability results. By contrast, the Bi-Bynel device showed slight changes (Supplementary Fig. 25), and the Bi-Barrier-Bynel device showed negligible change (Supplementary Fig. 26), which are consistent with their light-heat operational stability results. The results indicate that the LeTID of FACs perovskite film in Ag-Bynel device is effectively suppressed when Bi/Al$_2$O$_3$/parylene multiple-barrier is used. In addition to morphology change, it was also found that iodine ions diffused to the Ag electrode in the Ag-Bynel device, and the interdiffusion of Ag into perovskite was also observed (Supplementary Fig. 24). By contrast, the interdiffusion of Bi was not observed, and the outward migration of iodine species was significantly suppressed in the Bi-Barrier-Bynel device. This observation further confirms that Bi-Barrier-Bynel is a much more stable structure to suppress perovskite decomposition and improve PSCs' stability.

To further understand the mechanisms for the excellent stability of the Bi-Barrier-Bynel device, we measured the changes of the defect nature within the devices before and after the light-heat operational stability tests by admittance spectroscopy[40,41], and the Ag-Bynel device was taken as a reference (Supplementary Figs. 27 and 28). Dark capacitance-frequency ($C$-$f$) characteristics were measured on the devices at different $T$ without applying bias ($f$ ranges 100 to $1.5 \times 10^6$ Hz). It was found that the extracted trap energy level ($E_a$) of the Ag-Bynel device changed from 0.35 to 0.21 eV, which indicated the formation of massive shallow defects within the perovskite layer after aging[42,43]. This change should be associated with the decomposition of the perovskite layer, which is the reason why the device cannot work after operating for a while. Whereas for the Bi-Barrier-Bynel device, the

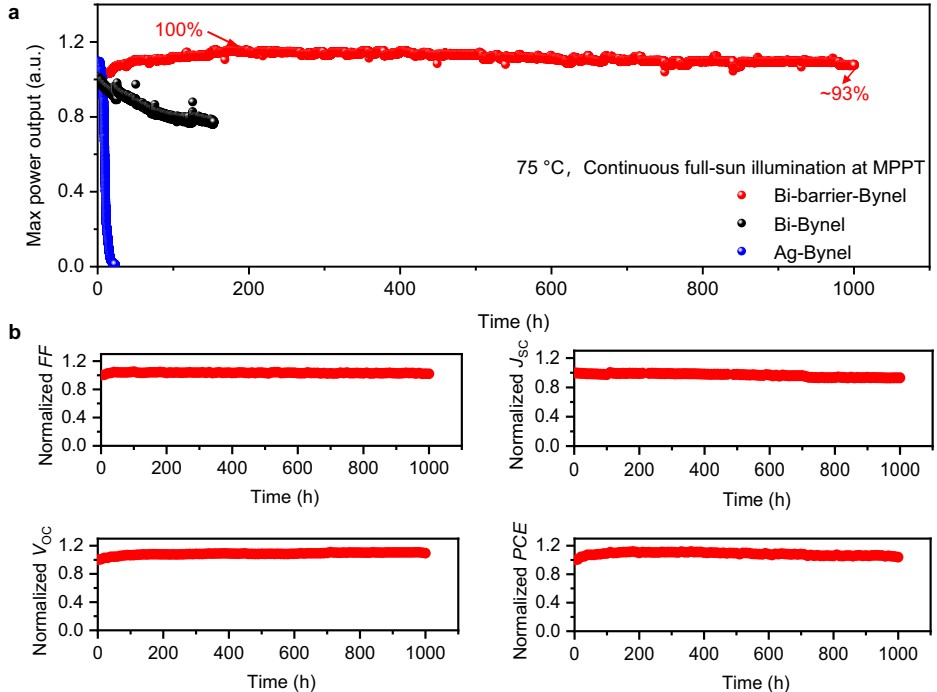

**Fig. 5 | LeTID stability test of the devices. a** Maximum power output evolution of devices monitored by MPP tracking at 75 °C under 1 sun equivalent white-light LED light illumination. The light intensity was calibrated to achieve the same $J_{SC}$ from the PSCs as for 1 sun AM1.5 G solar irradiation. **b** The evolution of all the parameters, including $J_{SC}$, $V_{OC}$, FF and PCE of the Bi-Barrier-Bynel device (aging condition: 1 sun equivalent white-light LED soaking, 75 °C). Source data are provided as a Source Data file.

corresponding trap energy level slightly changed from 0.36 to 0.33 eV (Supplementary Fig. 28), keeping a deep defect level, which indicates the decent integrity of the crystal structure of the perovskite layer after aging.

## Modulation mechanism of perovskite decomposition with multiple-barrier

To illustrate the modulation mechanism of perovskite decomposition in the assist of Bi electrode and Al₂O₃/parylene bilayer barrier more clearly, time-of-flight secondary ion mass spectroscopy (ToF-SIMS) was used to monitor the changes of the elemental distribution for each layer before and after light-heat aging at 85 °C for 100 h under $N_2$ atmosphere (Fig. 6a–d). For the Ag electrode-based sample, it is found that Ag penetrated into the deeper region of the perovskite layer, and the order of the magnitudes of the I⁻ ions intensity at the Ag electrode arrived at $10^5$, which was much higher than that of the fresh device with an I⁻ ions intensity of $10^3$ (Fig. 6c). These results indicated that there were massive I⁻ ions diffused outward toward Ag electrode and reacted with Ag to form AgI. Meanwhile, Ag also diffused into the perovskite layer and reacted with I⁻ ions. For the device with Bi electrode (Fig. 6b), the inward diffusion of Bi into perovskite was negligible, and the intensity of the outward diffusion of I⁻ ions decreased to the value of $10^4$ magnitude, which was much lower than the Ag case. In the Bi-based device, the migrated I⁻ ions will not be consumed by Bi, leading to the migrated I⁻ ions accumulating at the perovskite-$C_{60}$ interface and forming a charge barrier due to the chemically inert properties of Bi electrode[44]. These results are consistent with the long-term light-heat stability test of the Ag-no barrier and Bi-no barrier devices (Fig. 5a). And for the aged device with Bi electrode and bilayer barrier (Fig. 6d), the intensity of the migrated I⁻ ions near the electrode is $10^3$ magnitudes, which is similar to the value of the fresh device. This is because the degradation of perovskite was suppressed by the bilayer barrier and thus there were much fewer removable I⁻ ions compared to the

devices without barrier (Fig. 6e–g). It has been reported that I⁻ ions have the lowest ion migration activation energy among all mobile species in perovskite[45], making I⁻ the fastest migration species[46]. I⁻ could migrate throughout the perovskite absorber and into the charge transfer layer and react with the metal electrode, which is a recognized path of perovskite decomposition. During the aging process of the sample with an Ag electrode, a large amount of HI steam is produced (Fig. 2c), which could further diffuse into the electrode layer driven by the concentration gradient. Moreover, this process was accelerated by the consumption reaction between Ag and HI and the outgassing consumption of HI. Whereas because of the chemical inertness of the Bi electrode, the HI concentration differential between the Bi electrode and the perovskite layer is much smaller than that between the Ag electrode and the perovskite layer, which alleviates the diffusion of HI from the perovskite layer to the electrode layer. This is why the concentration of I⁻ at the Bi electrode is lower than that at the Ag electrode layer (Fig. 6a, b). For the sample with Bi electrode and bilayer, the difference in HI concentration between perovskite and electrode is smaller than that in the previous two cases due to the non-consumption (no reaction and no escape) of decomposition products. Thus, the Bi-bilayer barrier-based sample showed the lowest I⁻ concentration at the electrode interface (Fig. 6d).

To further gain insights into the modulation mechanism of perovskite decomposition with multiple-barrier, we examined the morphology and element change of the FACs perovskite film and the electrode in samples with different electrodes and barriers after light-heat aging by SEM-energy dispersive X-ray (EDX) and X-ray photoelectron spectroscopy (XPS) measurements. The samples for these measurements were obtained by removing the $C_{60}$ layer on the top of perovskite with chlorobenzene washing and adhesive tape treatment. For perovskite surface, the perovskite film in perovskite/$C_{60}$/Bi/Al₂O₃/parylene sample showed negligible morphology change after aging, and the negligible concentration of Bi element did not increase on the

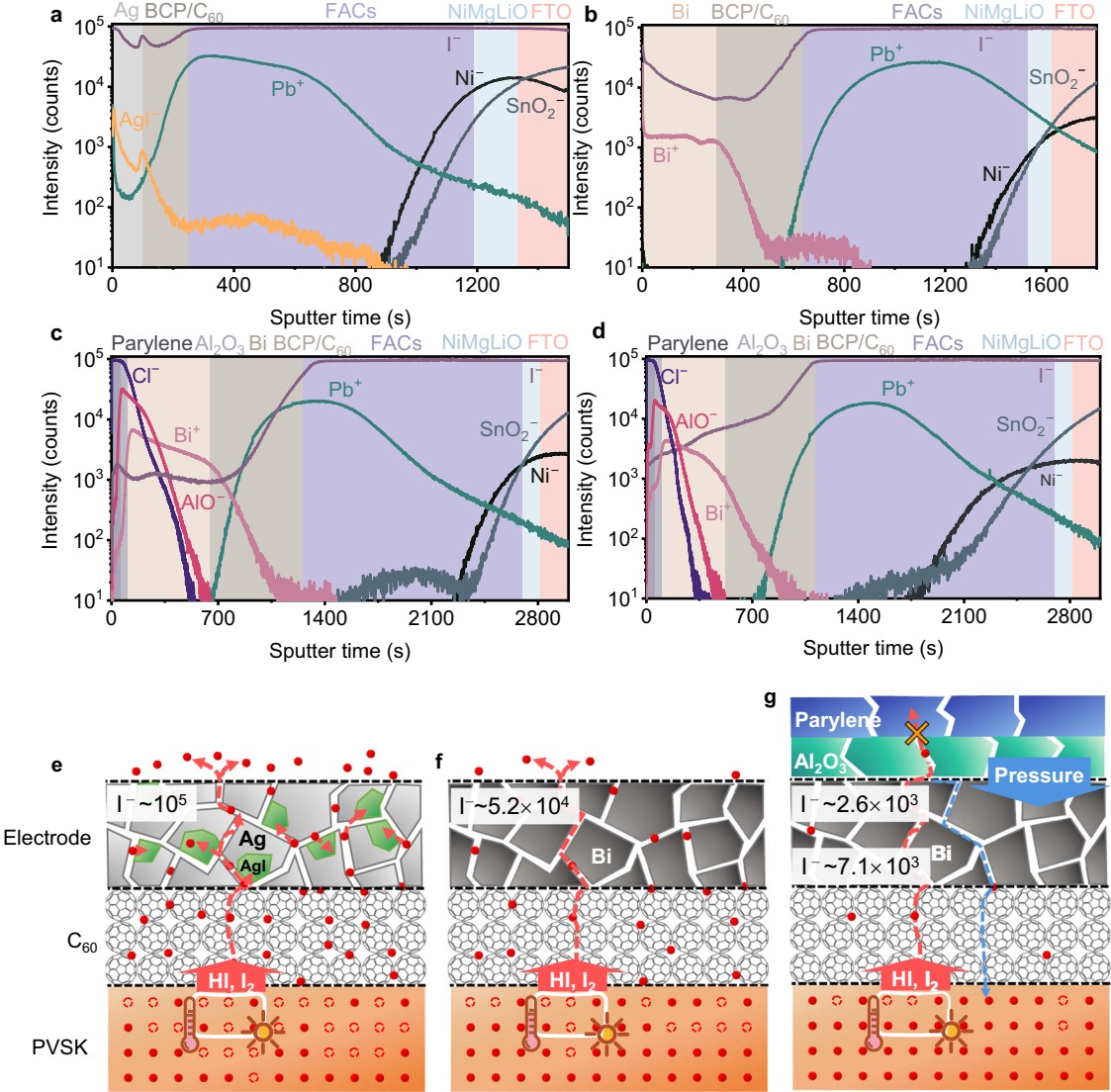

**Fig. 6 | Elemental distribution in aged devices and the corresponding schematic diagram of PSC degradation. a–d** ToF-SIMS elemental depth profiles of the aged device with Ag electrode, the aged device with Bi electrode (The thickness of Bi electrode is 200 nm here for the test), the fresh device with Bi electrode and bilayer barrier, the aged device with Bi electrode and bilayer barrier (aging condition: continuous 1 sun equivalent white-light LED illumination, in $N_2$ atmosphere, 85 °C, 100 h). **e–g** Schematic diagram of the iodine ions migration within the devices with three different architectures.

surface of perovskite after aging (Supplementary Fig. 29). However, the perovskite films of perovskite/$C_{60}$/Ag sample and perovskite/$C_{60}$/Bi sample without bilayer showed different degrees of degradation (Supplementary Figs. 30 and 31). This is because the decomposition of perovskite film is suppressed under the protection of the bilayer barrier, and meanwhile, Bi did not react with the I⁻ ions or diffused into the perovskite layer. The XPS results also confirm that there are negligible Bi elements on the surface of perovskite in Bi-based samples with and without barrier (Supplementary Fig. 32). For the inner interface of the electrode, both Bi layers in the corresponding samples with and without $Al_2O_3$/parylene barrier show negligible morphology change after aging. However, the Ag electrode showed severe degradation, and a massive Ag element was detected on the aged perovskite surface, which further confirms the interdiffusion of Ag into perovskite during aging. In addition, the relative content of iodine also showed a large variation in the aged Ag-based sample, and small change in the device with Bi electrode without barrier, and negligible change in the Bi-bilayer-based sample, respectively. This is because a large amount of iodine elements was consumed in the Ag-based sample due to the outgassing and Ag-halide reaction. For the Bi-based sample without

barrier, although the Bi electrode will not consume iodine, the unrestricted leakage of gaseous decomposition products of perovskite also induces the loss of iodine. Only the sample with Bi electrode and bilayer can maintain the iodine content after aging. The above results were consistent with the ToF-SIMS results of the corresponding devices. All these results confirm that the Bi/$Al_2O_3$/parylene multiple-barrier can help the device system reach relative benign equilibria, consequently largely slowing down the degradation rate of perovskite and devices.

Based on the above results, a dynamic analysis of the reversible decomposition reactions is further discussed from the viewpoint of reaction thermodynamics and kinetics. As shown in Fig. 6, the Ag electrode-based device without barrier can be regarded as an unclosed system or an open consumption system (OCS), and the forward decomposition rate is always much higher than the backward rate (Fig. 6e). For the Bi electrode-based device, but without $Al_2O_3$/parylene barrier, it is still an OCS, but without electrode consumption, the Bi electrode can block the outgassing of perovskite decomposition at a certain content (Fig. 6f). Taking advantage of the robust barrier capability of the $Al_2O_3$/parylene barrier, the Bi electrode-based device with

barrier can be recognized as a closed system or closed non-consumption system (CNS). In CNS, the forward decomposition rate will decrease and the reverse reaction rate will increase until the forward rate equals to reverse rate, and this system reaches a benign equilibrium and thus contributes to the best stability of the corresponding devices (Fig. 6g).

In this work, we demonstrated the modulation of perovskite decomposition with a robust $Bi/Al_2O_3$/parylene multiple-barrier to improve the light-heat operational stability of PSCs targeting LeTID. The inert Bi electrode can retard the reaction of perovskite decomposition products and metal electrode. The robust $Al_2O_3$/parylene bilayer encapsulation can greatly prevent the outgassing of the perovskite decomposition products. The $Bi/Al_2O_3$/parylene multiple-barrier leads to suppressing perovskite decomposition by allowing the decomposition reactions to reach benign equilibria, which can maintain the integrity of perovskite lattice structure at harsh light-heat conditions. Meanwhile, the bilayer barrier also protects the device against the ingress of water and oxygen. As a result, our FACs perovskite-based devices with the $Bi/Al_2O_3$/parylene multiple-barrier maintained 90% of their initial efficiencies after continuous operation at 45 °C for 5200 h and 93% of their initial efficiency after continuous operation at 75 °C for 1000 h under 1 sun equivalent white-light LED illumination. We anticipate that this work will open avenues for designing barriers to improve the operational stability of PSCs and solar modules.

## Methods

**Materials.** Nickel(II) acetylacetonate (95%) was purchased from TCI. Lithium acetate ($LiOOCCH_3 \cdot xH_2O$, 99.98%) was purchased from Alfa Aesar. Formamidinium iodide ($CH(NH_2)_2I$ (FAI), 99.5%) was purchased from Greatcellsolar. Lead (II) iodide (99.8% metals basis), cesium bromide (>99%), and bathocuproine (($C_{26}H_{20}N_2$, BCP), >99%) were purchased from Tokyo Chemical Industry Co., Ltd. Dimethyl Formamide (DMF, 99.8%), dimethyl sulfoxide (DMSO, 99.8%), chlorobenzene ($C_6H_5Cl$, 99.8%), magnesium acetate (($CH_3COO)_2Mg \cdot 4H_2O$, 99.99%), acetonitrile ($CH_3CN$, spectrographic grade) were purchased from Sigma-Aldrich. Isopropanol ($C_3H_8O$, analytically pure) was purchased from the Guoyao Group company. Ag, Cu, Au, and Bi, 99.99%, were purchased from Tanbang Co., Ltd. in Hebei province.

**Device fabrication.** FTO glasses (TEC-8, Nippon Sheet Glass Co., Japan) were cleaned through sequential ultrasonication for 20 min in a detergent solution, distilled water, alcohol, and acetone. Then, a p-type NiMgLiO was deposited on top of the FTO glass: a mixture solution of acetonitrile and ethanol (with 95:5 volume ratio, 30 mL) of nickel acetylacetonate (with magnesium acetate tetrahydrate and lithium acetate, and the mole atomic ratios of Ni:Mg:Li is 80:15:5, the total metal ion concentration is 0.02 mol $L^{-1}$) was sprayed by an air nozzle (with 0.2 mm caliber) onto the hot FTO glasses (570 °C). After spraying, the samples were further treated at 570 °C for another 40 min and then left to cool naturally. After cooling to RT, perovskite was then deposited by the anti-solvent method. The $FA_{0.85}Cs_{0.15}Pb(I_{0.95}Br_{0.05})_3$ perovskite films were deposited by using a precursor solution containing FAI, $PbI_2$, and CsBr in DMF:DMSO = 4:1 by volume. And it was doped by C343[47]. LiF (0.5 nm) was evaporated on the perovskite surface for further surface treatment. Then, $C_{60}$ (20 nm) layer was prepared by evaporation. In the final step, the BCP, Bi (20 nm), or Ag were deposited at high vacuum (less than $5 \times 10^{-4}$ Pa) while finely controlling the evaporation rate at 0.1, 0.1, and 0.1–0.5 Å $s^{-1}$, respectively. M-Bi (1 μm) was prepared by magnetron sputtering in a vacuum chamber ($\leq 5 \times 10^{-4}$ Pa). Then, $Al_2O_3$ film was deposited on the samples at RT by ALD using trimethylaluminium (TMA) and $H_2O$ as aluminum and oxygen precursors and $N_2$ as a purge gas. One complete ALD cycle consisted of 0.02 s of TMA, 12 s of $N_2$, 0.02 s of $H_2O$, and 12 s of $N_2$ pulses. The pressure in the ALD chamber was in the range of $10^{-1}$ mbar. The growth rate per cycle for ALD was

0.67 nm per cycle. Parylene was deposited by CVD with a pyrolysis temperature is 700 °C. The devices for encapsulation were first loaded into the CVD chamber, which was pumped to a pressure below 10 Pa. The precursor particles (Parylene C, 9 g) were maintained at a temperature of 120 °C for vaporization. Furthermore, the furnace tube was maintained at a temperature of 700 °C to pyrolyze the precursor to form monomers. During parylene deposition, the chamber was maintained at a pressure of approximately 10 Pa. The substrate temperature during deposition was approximately 40 °C. The thickness of the parylene encapsulation was approximately 1 μm. A cover glass was further attached to the top of each parylene-covered device to provide mechanical protection for the underneath layers. To achieve this, a glass sheet with a thickness of 1 mm was carefully sealed on top of the parylene film using Bynel adhesive. Then, the device was put in the laminating machine at 140 °C for 15 min to melt the Bynel adhesive to achieve cover encapsulation. It is noted that the cover glass with a small chamber at the site of the active area was designed to protect the devices from mechanical stress during the process of hot-press. In the end, the whole device was cast with epoxy resin, which is mixed with two-component glue in a certain proportion, then the biochemical crosslinking reaction was carried out at room temperature to realize curing and packaging without damaging the perovskite solar cell and at the same time to provide sufficient water and oxygen barrier ability.

**Characterization and measurement.** SEM images were obtained using a Nova NanoSEM 450 scanning electron microscope (FEI Co., Netherlands). The element distribution test was performed using a characterization technique combining SEM and EDS. By using SEM to select a specific area of the sample for EDS surface scanning analysis, the corresponding element distribution can be obtained. The instrument model is Nova NanoSEM 450. The crystal structure of the films was characterized using XRD with an Empyrean X-ray diffractometer with Cu Kα radiation (PANalytical B.V. Co., Netherlands). Volatile degradation traces were recorded using a quadrupole MS equipped with an electron multiplier detector (HAL3F501RC, Hiden Analytical). ToF-SIMS depth profiles were carried out using an IonToF ToF-SIMS 5 instrument (IONTOF Co., Germany). An XPS system (Thermo ESCALAB 250XI) was used to acquire the XPS spectra. Steady-state PL was measured using a Horiba Jobin Yvon system with an excitation laser beam at 532 nm. Admittance spectroscopy was obtained using a home-made combinatorial testing system mainly consisting of an impedance analyzer (Agilent E4980A LCR meter), a liquid nitrogen cryostat (Janis VPF-100), and a temperature controller (Lakeshore 325). Photovoltaic measurements employed a black mask with an aperture area of 1 $cm^2$ under standard AM1.5G simulated sunlight (Oriel Class AAA, XES-160S1, SAN-EI ELECTRIC CO., LTD., Japan) with a source meter (Keithley 2601 B), and the simulated light intensity was calibrated with a silicon photodiode. The light-heat operational stability at 75 °C was measured in the air atmosphere with a multi-channel automated stability testing system (91PVKSolar Co., Ltd). It should be noted that a water-cooling system is employed to guarantee the long-term constant light output (<2% degradation for 10,000 h) of the light source.

### Reporting summary
Further information on research design is available in the Nature Portfolio Reporting Summary linked to this article.

## Data availability
The data that support the findings of this study are available in the Supplementary Information/Source Data file. Source data are provided with this paper.

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

## Acknowledgements

This work was financially supported by the National Key Research and Development Project funding from the Ministry of Science and Technology of China (2021YFB3800104), the National Natural Science Foundation of China (52002140, U20A20252), the Young Elite Scientists Sponsorship Program by CAST, the Self-determined and Innovative Research Funds of HUST (2020kfyXJJS008), the Natural Science Foundation of Hubei Province (2022CFA093), and the Innovation Project of Optics Valley Laboratory (OVL2021BG008). G.T., L.K.O. and Y.B.Q. acknowledge the funding support from the Energy Materials and Surface Sciences Unit of the Okinawa Institute of Science and Technology Graduate University, the OIST R&D Cluster Research Program, and the OIST Proof of Concept (POC) Program. The authors thank the Analytical and Testing Center of Huazhong University Science and Technology for the sample measurements.

## Author contributions

Z.Liu, W.C., and Y.B.Q. designed and directed the study. J.Z. and P.Y. conceived and performed the device fabrication work. J.Z. and Z.Liu optimized the thin-film barriers. P.Y. optimized the sputtering process of Bi electrode. G.T. and L.K.O. carried out the mass spectroscopy measurements and data analyses. J.Z. and Rui.C. carried out the cross-sectional SEM characterization of the PSC. W.C., J.Z., Ruij.C. and Z.Lan designed and assembled the parylene deposition system. J.Z., P.Y., Rui.C., H.X.W., F.M.R., S.W.L., J.N.W. and L.K.O. participated in the experiment. All authors contributed to the discussions. J.Z., G.T., Z.Liu, W.C. and Y.B.Q. wrote the draft manuscript with input from all authors. All authors revised the paper.

## Competing interests

The authors declare no competing interests.
