## [Peer Review File · Nature Communications]

Modulation of perovskite degradation with multiple-barrier for light-heat stable perovskite solar cellsEditorial Note: This manuscript has been previously reviewed at another journal that is not operating a transparent peer review scheme. This document only contains reviewer comments and rebuttal letters for versions considered at *Nature Communications*. Mentions of the other journal have been redacted.

Reviewer #1 (Remarks to the Author):

I have reviewed an earlier version of this manuscript for [redacted]. The authors have addressed most of the concerns of the referees. They have provided a lengthy answer to my criticism of limited novelty and impact (I have provided numerous references on prior work in my earlier report). Their response only partially convinced me. In the manuscript, the authors refer to their previous paper, which already discusses in detail the use of Bi as a barrier in perovskite solar cells [<https://doi.org/10.1038/s41467-019-09167-0>] and say: "... we extend our discovery of the chemically inert bismuth(Bi) as a 20 nm-thick interfacial layer to an 1 μm -thick back contact electrode by optimizing its deposition process." The added Parylene/Al₂O₃ is well established. I would leave it to the editor to decide if this level of innovation is within the scope of this journal.

Reviewer #3 (Remarks to the Author):

After carefully reviewed both the original manuscript, revised manuscript and the reviewers' comments, I find that the authors have addressed most of the concerns by Reviewer 2. However, I also agree with the novelty issue raised by Reviewer 1. The stability results in this work is definitely important to the perovskite solar cells field. However, I feel the work is more technical rather than science. The authors bringing in a few well known strategies in encapsulation together. Besides the novelty issue, there are a couple of questions needing further clarifications.

1. In Figure S2, the authors reported the reproducibility of the devices upon bilayer encapsulation but didn't give the number of devices tested.

2. For the devices that sent for stability certification, the authors used bynel and epoxy for further encapsulation. Why? I wonder if other devices with bynel and epoxy encapsulation also have good stability.

Response to Reviewer #1

Reviewer #1 (Remarks to the Author):

I have reviewed an earlier version of this manuscript for [redacted]. The authors have addressed most of the concerns of the referees. They have provided a lengthy answer to my criticism of limited novelty and impact (I have provided numerous references on prior work in my earlier report). Their response only partially convinced me. In the manuscript, the authors refer to their previous paper, which already discusses in detail the use of Bi as a barrier in perovskite solar cells [<https://doi.org/10.1038/s41467-019-09167-0>] and say: "... we extend our discovery of the chemically inert bismuth(Bi) as a 20 nm-thick interfacial layer to an 1 μm -thick back contact electrode by optimizing its deposition process." The added Parylene/Al₂O₃ is well established. I would leave it to the editor to decide if this level of innovation is within the scope of this journal.

→ We appreciate the reviewer's positive comment about our work and the help to improve the clarification about novelty. Taking advantage of this opportunity, we would like to further explain the innovation of our work.

In terms of electrode, we report an intrinsic stable metal electrode scheme for PSCs. a)

Intrinsic stability of electrode: Although we have used Bi as an interface barrier to prevent the migration of halide ions and metal to a certain extent in our previous work (*Nat. Comms* 2019, 10, 1161). The problem with such barrier design is that the corrosion of metal electrodes is not solved fundamentally. The active metals such as Ag were still used on the top of the Bi barrier, which can still consume perovskite decomposition components and cause perovskite decomposition within the system created by encapsulation. Here, we upgraded the previous 20 nm-thick Bi barrier to an inert 1 μm -thick Bi electrode and avoided the use of an active metal electrode. The Bi is used as a metal electrode instead of Ag, Cu et al. active metal, and the Bi is chemically inert, which will not react with the perovskite composition. The instability caused by the consumption of perovskite by the active metal electrode is solved fundamentally. b) **Improved conductivity and compatibility with the fragile perovskite.** We solved the low conductivity problem of the thermally evaporated 1 μm -thick Bi electrode and the damage of the sputtering process on perovskite by the combination of

thermally evaporated 20 nm Bi barrier and sputtered 1 μm -thick Bi electrode. Such combination not only improved the conductivity and morphological compactness of the electrodes, but also increased the electrode deposition rate and utilization rate of raw materials, making it beneficial for industrial applications. With the protection of thermally evaporated 20 nm Bi barrier, such electrode scheme is also compatible with the fragile perovskite underlayer.

In terms of the modulation of perovskite decomposition, we integrated the intrinsic electrode with a scalable and robust Al_2O_3 /parylene barrier encapsulation scheme from the perspective of thermodynamics. We used the Al_2O_3 /parylene bilayer barriers to create a more compact barrier to form a compact and enclosed system, which not only helps to retard the penetration of water/oxygen in ambient air but also suppresses the removal of perovskite decomposition byproducts. Once the degradation occurs, the concentration of the perovskite decomposition byproducts can reach a high level to make the decomposition reactions of perovskite reach benign equilibriums.

Based on the above innovations, we can achieve highly stable PSCs under harsh light-heat aging conditions. We think the multiple-barrier strategy is vital for the practical application considering its effectiveness in solving the stability problem of PSCs.

Response to Reviewer #2

Reviewer #3 (Remarks to the Author):

After carefully reviewed both the original manuscript, revised manuscript and the reviewers' comments, I find that the authors have addressed most of the concerns by Reviewer 2. However, I also agree with the novelty issue raised by Reviewer 1. The stability results in this work is definitely important to the perovskite solar cells field. However, I feel the work is more technical rather than science. The authors bringing in a few well known strategies in encapsulation together. Besides the novelty issue, there are a couple of questions needing further clarifications.

→ We appreciate the reviewer's positive comment about our work. About the novelty, we want to emphasize that this work not only develops a robust electrode+encapsulation scheme but also introduces a holistic consideration of PSCs' stability and the modulation of perovskite decomposition from the perspective of thermodynamics. Our work reports the findings from the perspective of thermodynamics and uses in situ X-ray diffraction (XRD) and in situ mass spectroscopy (MS) characterization techniques to fully demonstrate the feasibility of the inert electrode (does not react with perovskite) + barrier (create tight enclosure) strategy to solve one of the most challenging stability issues in this field, i.e., light and elevated temperature induced degradation (LeTID). Moreover, different from previous works using expensive gold electrodes or ITO electrodes, we used inexpensive and inert metal with good process reliability, i.e., bismuth (Bi) as an electrode as well as a metal barrier. The multiple-barrier strategy used in this work is thus novel when compared to previous barrier strategies to boost the LeTID of PSCs. We think the multiple-barrier strategy is vital for the practical application considering its effectiveness in solving the stability problem of perovskite solar cells. Therefore, we strongly insist on our novelty.

1. In Figure S2, the authors reported the reproducibility of the devices upon bilayer encapsulation but didn't give the number of devices tested.

→ We would like to thank the reviewer for the comments. As shown in revised Figure S2, the diagram mode has been modified with the inclusion of data points. In total, there are 18 devices with

36 data points, consisting of two points for forward and reverse scans of each device. We have clarified this point in the revised Supplementary Information. **For the corresponding revisions, please refer to the revised Supplementary Information, Page 8.**

2. For the devices that sent for stability certification, the authors used bynel and epoxy for further encapsulation. Why? I wonder if other devices with bynel and epoxy encapsulation also have good stability.

→ We would like to thank the reviewer for the comments. The use of Bynel and epoxy is intended to provide external mechanical protection and also retard the penetration of H₂O and O₂ in ambient air. We have compared the stability of the Bynel and epoxy encapsulated devices with Ag electrode, Bi electrode, and Bi electrode+Al₂O₃/parylene barrier. For a mild aging condition, 45 °C + one Sun, MPPT, all of the devices showed relatively good stability (Supplementary Figure 22). But once the light-heat aging condition, i.e., 75 °C + one Sun, MPPT, was adopted, the former two devices showed apparent degradation, especially for the Bynel-Ag electrode. This may be due to the Ag-induced degradation being triggered and the Bynel+epoxy encapsulation is also not robust enough to retard the removal of perovskite degradation byproducts and the penetration of H₂O and Oxygen in ambient air. For the case of Bi-Bynel device, the Bynel+epoxy encapsulation is not robust enough to retard the outgassing of perovskite degradation products and the penetration of H₂O and Oxygen, resulting in more serious degradation of perovskite and poorer high-temperature device stability when compared with the device with Bi electrode+Al₂O₃/parylene multiple barriers.